# Microangiopathy in Naifold Videocapillaroscopy and Its Relations to sE- Selectin, Endothelin-1, and hsCRP as Putative Endothelium Dysfunction Markers among Adolescents with Raynaud’s Phenomenon

**DOI:** 10.3390/jcm8050567

**Published:** 2019-04-26

**Authors:** Stanislaw Gorski, Marta Bartnicka, Anna Citko, Beata Żelazowska-Rutkowska, Konrad Jablonski, Anna Gorska

**Affiliations:** 1Department of Medical Education, Jagiellonian University Medical College, 31-530 Krakow, Poland; konrad.jablonski@uj.edu.pl; 2Department of Family Medicine, Medical University of Bialystok, 15-054 Bialystok, Poland; martawozniak@o2.pl (M.B.); agorska50@wp.pl (A.G.); 3Outpatient Clinic, Bialystok Children’s Clinical Hospital of L. Zamenhof, Medical University of Bialystok, 15-274 Bialystok, Poland; anka234@gmail.com; 4Department of Pediatric Laboratory Diagnostics, Medical University of Bialystok, 15-276 Bialystok, Poland; beata.zelazowska@umb.edu.pl; 5Department of Pediatrics, Rheumatology, Immunology, and Metabolic Bone Diseases, Rheumatology Outpatient Clinic, Medical University of Bialystok, 15-276 Bialystok, Poland

**Keywords:** Raynaud’s phenomenon, videocapillaroscopy, biomarkers, lipid profile, endothelin dysfunction, children, adolescents, atherosclerosis

## Abstract

The aim of this study was to analyze the relationship between the qualitative abnormalities on nailfold videocapillaroscopy (NVC), and the concentrations of selected biomarkers (sE-selectin, endothelin-1, high-sensitivity c-reactive protein (hsCRP)) and lipid metabolism parameters in children and adolescents with Raynaud’s phenomenon (RP). Raynaud’s phenomenon, to assess whether nailfold capillary changes may reflect the degree of systemic blood vessel abnormalities. The study group included 66 patients (34 undifferentiated—uRP and 32 secondary—sRP) aged 6–19 years and the control group. In both groups, NVC was performed and the selected biomarkers were measured (sE-selectin, endothelin-1, hsCRP) and lipid profile. Endothelin-1, sE-selectin and hsCRP concentrations in patients from both RP groups were significantly higher; concentration of HDL fraction was significantly lower compared with the control group. The analysis of multiple linear regression demonstrated that megacapillaries most strongly determine the sE-selectin value (*p* = 0.04) and hsCRP (*p* = 0.03). Both the total cholesterol and low-density lipoprotein (LDL) fraction concentrations were determined by the presence of avascular areas (*p* = 0.02). In conclusion, specific pathologic NVC changes were associated with higher endothelial damage biomarkers concentration and adverse changes in the lipid profile.

## 1. Introduction

Microcirculation is an important, although still relatively insufficiently known component of the cardiovascular system, both in a structural and functional respect [1,2,3]. 

Clinical manifestation of peripheral microcirculation disorders is a Raynaud’s phenomenon (RP), which may precede autoimmune connective tissue diseases (CTDs) many years, or accompany them [4].

RP consists of three phases, namely the ischemic phase, cyanotic phase and erythema phase. The vasoconstriction of pre-capillary sphincters of the digital arteries and arterioles induced by cold and/or emotional stress determines the initial marked pallor (ischemic phase); then pre-capillary sphincters relax due to hypoxia and accumulation of anaerobic products, determining a hyper-flow of blood in ischemic areas, with a rapid hemoglobin desaturation (cyanotic phase); finally, a pre and a post-capillary sphincter dilatation induces a reactive, often painful, hyperemia (erythema phase). 

An established method for the evaluation of structural abnormalities of the microcirculation is the nailfold videocapillaroscopy (NVC). The videocapillaroscopic examination is non-invasive and inexpensive, thus allowing multiple re-testing. Meanwhile, the computerized analysis of images of the microcirculation and the possibility of data storage enable assessment of the dynamics of changes taking place over time [5,6,7].

While capillaroscopic vascular assessment can be carried out at many points in the human body, the most common studies are the microcirculation of the nailfold-nailfold videocapillaroscopy. This method informs the researcher about the blood supply of tissues (shape, size, number and filling of capillaries) and the degree of nourishment of the skin (background image or fields with no blood vessels)

NVC is also a safe tool to examine the microcirculation in children and adolescents with CTDs. 

NVC in rheumatologists’ practice has a dual use. First, it has a role in the differential diagnosis of patients with RP. Second, it may have a role in the prediction of clinical complications in CTDs [8]. 

Furthermore, some researchers suggest that local microangiopathy found on NVC in RP patients with/without CTDs also reflect disorders in the systemic microcirculation, including cardiovascular, pulmonary or cerebral microcirculation [9,10,11].

The monitoring of morphological abnormalities in capillary vessels using NVC, particularly in patients with long-standing Raynaud’s phenomenon, may also provide an indirect indication of the presence of activation and/or dysfunction endothelial within the vessels of the microcirculation [12,13]. 

Hence, 2 hypotheses seemed worth checking out. First of all, the structural microcirculation abnormalities found in NVC may be the symptoms of subclinical vasculitis in the course of RP. Secondly, that NVC in combination with endothelial biomarkers may be a useful diagnostic tool to identify preclinical prognosticators of endothelial dysfunction.

Endothelial dysfunction is the main trigger for the release of endothelin-1. Endothelin-1 is the strongest vasoconstrictor, and in the RP, its release may stimulate, among others, low temperature, shear stress or hypoxia [14]. In the pathomechanism of the RP, the role of ET-1 is still controversial, although many researchers suggest that its increase in serum in patients with RP primarily indicates the degree of damage to the vascular endothelium [15,16].

Like ET-1, E-selectin is specific for endothelial cells and only produced by them in contrast to other adhesion molecules [12,17]. E-selectin at the early stage of endothelial activation mediates the adhesion of neutrophils, monocytes, and T lymphocytes to the vascular wall, which may be associated with increased permeability and subclinical vasculitis [18]. Hebbar et al. found increased expression of E-selectin on endothelial cells in patients with Raynaud’s syndrome, who were followed by systemic sclerosis (SSc) development [19]. The literature emphasizes that low-density lipoprotein (LDL), after oxygenation, has an adverse effect on vasodilatation of vessels by stimulating the secretion of ET-1. It is also suggested that HDL cholesterol inhibits the expression of E-selectin and mediating of monocytes. What is more, long-lasting endothelial activation increases the passage of lipoproteins into the endothelial space, their retention, and modification [20,21].

The importance of the release of the biochemical markers of endothelial damage in patients with many years of Raynaud’s phenomenon history and capillaroscopic signs of microangiopathy can be considered in the context of preterm preclinical atherosclerosis [8]. The studies on the adult population of patients with CTD with or without RP unequivocally demonstrate an increased risk of cardiovascular disease, as compared with the healthy population [22,23]. Many researchers suggest that high-sensitivity c-reactive protein (hsCRP) is a putative marker of inflammation associated with atherosclerosis at its preclinical stage, stronger, and perhaps even more objective than the classic lipid parameters [24,25]. 

Few published biophysical and biochemical studies on microcirculation damage in patients with juvenile autoimmune diseases are available and they mainly concern patients with juvenile idiopathic arthritis (JIA) [25]. Even less information assessing the possible relationship between capillaroscopic changes in microcirculation and endothelial activation markers in juvenile Raynaud’s syndrome without CTD manifestations is available [26]. 

It seems, therefore, a reasonable attempt to assess the usefulness of NVC in young patients with RP, on the one hand, to assess the qualitative abnormalities in the nailfold videocapillaroscopy to monitor the course of RP, and on the other, the analysis of the relationship between structural changes in NVC and the concentration of endothelial damage biomarkers (sE-selectin, endothelin-1), lipid fractions and hsCRP in children and adolescents with the Raynaud phenomenon in the aspect of the risk assessment of subclinical vasculitis with possible inflammatory endothelial dysfunction.

## 2. Patients and Methods

### 2.1. Patients

Patients with Raynaud’s phenomenon (RP), treated at the Rheumatology Outpatient Clinic, University Paediatric Teaching Hospital, in whom NVC was routinely performed, served as the basis for selection of the study group. The degree of abnormalities on NVC, clinical assessment, RP history, and serological status enabled distinguishing between two RP subtypes:

Undifferentiated RP *(uRP)*: 34 patients with nonspecific NVC changes and without signs of systemic connective tissue disease (CTD) [5,27].

Secondary RP (*sRP)*: 32 patients with specific pathological NVC changes and diagnosed CTD or incomplete diagnosis—undifferentiated connective tissue diseases (UCTDs) [28,29].

In all, the study group included 66 patients (42 girls and 24 boys) aged 6–19 years (mean age 14.5 ± 4.52). 

Patients were divided into 2 groups. Group I: uRP-34 patients (mean age 13.07 ± 3.85), among them 25 (76.4%) females and 9 (23.6%) males. II group (sRP) consisted of 32 patients (mean age 15.29 ± 4.52), 20 (65.7%) females and 12 (34.3%) males.

The mean duration of Raynaud’s phenomenon was 4.82 ± 3.21 years.

The control group included 20 healthy children matched for age and gender: (mean age 14.73 ± 3.10), 15 girls (75%) and 5 boys (25%) coming for control visits to the Rheumatology Outpatient Clinic, without any cardiovascular burden and after ruling out infection on the basis of medical examination.

### 2.2. Methods

#### 2.2.1. Nailfold Capillaroscopy

All the patients had a nailfold capillaroscopy with the use of a ZEISS STEMI 2000 capillaroscope. An optical microscope was connected to a color digital camera and a personal computer.

The images taken during the examination were recorded on the computer’s hard disc drive (HDD) and analyzed by the same experienced investigator, pediatric rheumatologist (AG). 

The second investigator (MB) performed measurements of the number of capillaries on the capillaroscopic images recorded on computer’s HDD by means of the NIS-ELEMENTS D2.30 computer program, Nikon Corporation, Tokyo, Japan. 

The second experimenter was blinded to the children’s condition

The nailfold videocapillaroscopy (NVC) examinations were performed after at least 20 min. of acclimatization in a room with a constant temperature of 20–22 °C. Onto the examined finger cedar oil was applied to ensure better light penetration. The light was shed at an angle of 45° from a ZEISS KLD 1500 LCD lamp.

#### 2.2.2. Qualitative Measurements

The structure of the capillaries was assessed by examination of the nailfold bed capillaries in 200× magnification in all fingers of both hands (except thumbs). The capillary patterns were described as:non-specific pattern with a presence in at least two fingers: meandering and crossed capillaries, non-homogeneous distribution or size of loops (irregular arrangement), focal distribution of capillary hemorrhages, capillary spasm, widening of the afferent, apical and efferent parts of a loop, prominent subpapillary plexus [27].specific pathological (scleroderma pattern) with a presence of giant capillaries (3–4 times wider than the neighboring ones), frequent capillary hemorrhages, loss of capillaries (decrease of the capillary density <6 capillaries/linear mm), mild disorganization of the capillary architecture, the presence of avascular areas [29].

The number of tortuous and irregular vascular loops above two per one finger were regarded as an abnormal pattern.

Exemplary capillaroscopic patterns are shown in Figure 1, Figure 2 and Figure 3.

#### 2.2.3. Analytical Methods

The blood for analysis was collected on an empty stomach, during a control visit to the Rheumatology Outpatient Clinic or during hospitalization. The sera obtained after centrifugation were stored at a temperature of −80 °C:sE-selectin concentration was determined by the ELISA immunoenzymatic test method, using R&D Systems Quantikine kit, according to the test manufacturer’s instructions. The intensity of the color reaction was determined at 450 nm wavelength. The test sensitivity was 0.027 ng/mL;endothelin-1 concentrations were determined by the ELISA method, using a kit produced by IBL International. The degree of staining was measured at 450 nm wavelength and was proportional to serum ET-1 concentration. The measurement range was from 0.78 to 100 pg/mL and the test sensitivity was 0.23 pg/mL;hsCRP concentration was determined by the immunoturbidimetric method with Roche Tina-quant CRP HS reagent, using HITACHI 912 biochemical analyzer. The normal range was 0–0.5 mg/dL.

Lipid profile was determined by the enzymatic method using ready-to-use kits (Cormay, Poland). The biochemical tests were performed at the Division of Paediatric Laboratory Diagnostics, University Paediatric Teaching Hospital.

#### 2.2.4. The Statistical Analysis

The statistical analysis started with a material and logical review of collected data. In the first stage, conformity and distribution of the studied continuous variables were verified against the Gauss distribution. The hypothesis on the normality of the distribution was discarded. For this reason, the non-parametric U Mann–Whitney test was used in the analyses (two compared groups). For comparison of three or more independent groups, Kruskal–Wallis test which is comparable to one-way ANOVA is used. The relationships between qualitative variables were evaluated with the chi-squared test. The relationship between two quantitative variables was analyzed using multiple linear regression models. The assumed statistical confidence level was *p* < 0.05. The statistical analysis was performed using the STATA/1.C 12.1 application manufactured by StataCorp LP, College Station, TX, USA.

#### 2.2.5. Ethical Issues

The study was approved by the local Bioethics Committee of the Medical University of Bialystok, nr R-I-002/136/2015.

## 3. Results

Secondary RP (sRP) was diagnosed in 32 (47.15%) of the studied subjects (9–localized scleroderma-morphea or linear), 5–limited systemic sclerosis, 4–juvenile idiopathic arthritis, 3–juvenile dermatomyositis, 11–UCTDs. The remaining 34 (52.85%) patients were diagnosed with undifferentiated RP. 

### 3.1. Analysis of Capillaroscopic Parameters

Qualitative analysis of capillary parameters in the group of patients with RP and in the control group is shown in Table 1.

Analysis of qualitative capillary parameters in the group of patients with RP in relation to gender showed a significantly more frequent presence of meandering capillaries and giant capillaries among girls, *p* = 0.017 and *p* = 0.018, respectively; while the presence of neoangiogenesis was significantly more frequent in boys: *p* < 0.001. The results are shown in Table 2.

Analysis of qualitative capillary parameters in the group of patients with RP in relation to age showed a significantly more frequent presence irregular arrangement, tortuous capillaries, meandering capillaries and giant capillaries in the group of patients over 15 y.o. *p* = 0.017 i *p* = 0.018, *p* = 0.005 i *p* = 0.011, respectively. The results are presented in Table 3.

The statistically significant differences between group with sRP and in the control group were found concerning reduced density of capillaries (*p* = 0.002), irregular arrangement (*p* < 0.001), number of meandering capillaries (*p* = 0.03), presence of giant capillaries (megacapillaries) (*p* < 0.001), microhemorrhages (*p* = 0.015) and avascular areas (*p* < 0.003) (Table 4).

In the group of patients with uRP, significant differences compared to the control group were found only in the presence of irregular arrangement (*p* < 0.0001) and presence of megacapillaries (*p* < 0.001). No significant differences between the uRP group and the control group in respect of the number of tortuous capillaries and spastic capillaries were found. It should be stressed that no megacapillaries and avascular areas were detected in the control group (Table 5).

### 3.2. Analysis of Biochemical Tests

The clinical and biochemical parameters in the studied subjects with undifferentiated and secondary Raynaud’s phenomenon and in the control group are shown in Table 6.

The mean duration of Raynaud’s phenomenon was not different between the URP and SRP groups. A statistically significantly higher hsCRP concentration was demonstrated in URP and SRP patients compared with the control group (*p* < 0.001). The concentrations of endothelin-1 and sE-selectin in patients in both groups were significantly higher compared with the control group (*p* = 0.0134, *p* < 0.001, respectively) (Table 6). 

Analysis of biochemical parameters in relation to gender revealed significantly elevated levels of LDL and sE-selectin in boys, *p* = 0.018, 0.011, respectively; while in girls significantly higher concentrations of hsCRP (*p* = 0.017) were present. However, depending on the age, significantly higher concentrations of LDL, TG, hsCRP, ET-1 in the group of older / over 15 years / respectively were found: *p* = 0.002, *p* = 0.022, *p* = 0.030 and *p* = 0.016. In contrast, HDL concentration in the group above 15 years was significantly lower (*p* < 0.001) (Table 7 and Table 8).

On the other hand, when comparing the concentrations of the analyzed biomarkers between both RP groups, a statistically significant difference was found only in respect of sE-selectin concentration (Figure 4). The analysis of the lipid profile revealed a significantly lower HDL fraction concentration in RP patients compared with the control group (*p* < 0.001). The remaining lipid parameters were not differing significantly between the study groups. 

The U-Mann–Whitney test were used to compare the following groups: uRP (*n* = 34), sRP (*n* = 32), control group (*n* = 20). While comparing the groups, the results of endothelin-1 and sE-selectin measurements were taken into consideration: Endothelin-1 concentration in the uRP group (median = 3.46, Q25 = 2.34, Q75 = 4.41), endothelin-1 concentration in the sRP group (median = 2.97, Q25 = 2.105, Q75 = 4.385), endothelin-1 concentration in the control group (median = 1.93, Q25 = 1.22, Q75 = 3.01), sE-Selectin concentration in the uRP group (median = 35.92, Q25 = 31.2, Q75 = 42.25), sE-selectin concentration in the sRP group (median = 44.15, Q25 = 37.46, Q75 = 58.34), sE-selectin concentration in the control group (median = 22.315, Q25 = 17.225, Q75 = 28.295).

The analysis of multiple linear regression demonstrated that megacapillaries were the capillaroscopic parameters most strongly determining the sE-selectin value (Table 9).

On the other hand, the presence of avascular areas and microhemorrhages were the capillaroscopic parameters most strongly determining the hsCRP value in the group with Raynaud’s phenomenon in multiple regression analysis (Table 10).

None of the analyzed capillaroscopic parameters exerted any significant effect on the endothelin value.

On the other hand, multiple regression analysis of biochemical parameters shown a significant correlation between hsCRP and endothelin-1 (*R*² = 0.25, β = 0.31, *p* = 0.049).

Lipid metabolism parameters were also assessed. Both the total cholesterol and LDL fraction concentrations were determined by the presence of avascular areas: *R*² = 0.23, β = 0.34, *p* = 0.02, *R*² = 0.19, β = 0.36, *p* = 0.02, respectively. 

## 4. Discussion

In the opinion of some authors, the abnormal capillaroscopic pattern can be seen in healthy children in as much as 15–37% of cases [30,31]. In our studies, the abnormal capillaroscopic results in the control group concerned only the presence of tortuous capillaries (35%), meandering capillaries (20%) and spastic loops (24%). It should also be stressed that in the control group no vessel reduction, avascular areas or megacapillaries were found on NVC. Other researchers also revealed no capillaroscopic images of scleroderma pattern in their control group [5,27,30].

Ingegnoli and Herrick, based on the review of many studies, found that the capillaroscopic image of the nail shaft does not differ in healthy boys and girls. The tendency to increase the density of capillaries with age and is part of the maturation process, while the number of non-specific abnormalities in the vascular loop image increases with age [32,33]. The results of our research in adolescents with RP over 15 years, showed significantly more frequent abnormalities of the microcirculation architecture in NVC (irregular arrangement, tortuous and meandering capillaries, respectively: *p* = 0.014, *p* = 022, *p* = 0.005) compared to the group of subjects under 15 years. 

In general, the analysis of capillaroscopic parameters comparing males and females did not show significant differences in the microcirculation architecture. The significantly higher presence of giant capillaries (*p* = 0.017) in female patients can be explained by an increased proportion of sRP compared to the male gender (females-65.7% versus males-34.3%)

Concerning absence of established common terminology of capillaroscopic patterns in children and adolescents and limited data on among this age group, these finding can be interesting both for further research as well as practical application in pediatric rheumatologists’ practice.

A very interesting result of our research is the finding of significantly more frequent "scleroderma pattern" images in NVC in the patients with the RP. Our studies revealed a significantly more frequent occurrence of giant capillaries (*p* = 0.008), microhemorrhages (*p* = 0.031) and significantly reduced number of capillaries in the patients with Raynaud’s syndrome compared to healthy children in the control group (*p* = 0.015), which may indicate a subclinical process of vasculitis and the threat of organ-related complications, even in CTDs with low inflammatory activity. 

It should be emphasized that the abnormal capillaroscopic results were more frequent significantly different in patients with sRP than uRP compared to the control group.

No significant differences were found concerning the presence of tortuous capillaries and spastic loops compared to the control group. 

Studies of the adult population suggest that the presence of an abnormal image of the "scleroderma pattern" is a predictor of CTD in the future, despite the absence of clinical symptoms at the time of the study [34]. However, similar reports on children and adolescents are very limited [30,35]. In clinical practice, making a diagnosis in young patients with the Raynaud syndrome is difficult, hence the need to regularly repeat NVC in this group of patients. This is also related to the fact that the image of the vascular loops in patients with the secondary symptom of Raynaud sRP can undergo rapid progression. According to reports, even 15–20% of patients with the "scleroderma pattern" developed connective tissue diseases in 2–10 years, although initially, they did not meet Raynaud’s criteria [31,36].

The relationship between structural capillaroscopic abnormalities and endothelial dysfunction biomarkers seems interesting, although poorly elucidated.

The relationship between morphological abnormalities of capillaries, biomarkers of activation and inflammatory endothelial dysfunction in autoimmune diseases occurring with or without RP is interesting, though little understood, [1,32,37]. Activation of the endothelium is the first response of the vascular wall to mechanical factors (including disturbed flow), chemical (hypoxia, increased concentration of oxygen free radicals), biochemical (an increase of LDL concentration, cytokines) or biological (including antibodies). Regardless of the type of endothelial damage factors, the consequence is vasoconstriction associated with the release of Endothelin -1 in the first place [38,39].

The studies of endothelin-1 in RP patients conducted as yet, concerned its participation in the pathomechanism of RP. It has been demonstrated that its secretion is stimulated by low temperature, pressure increase or vascular wall stretching, angiotensin II, interleukin 1 (IL-1) and hypoxia [15,38].

A significant relationship between the presence of microangiopathy on NVC and higher serum ET-1 levels was demonstrated in Raynaud’s phenomenon both in children [26] and adults [40,41].

This is confirmed by the results of our studies in which serum ET-1 concentration was significantly higher in both uRP and sRP compared to the control group (*p* = 0.0134). Moreover, the higher ET-1 concentration in the sRP group compared to uRP, although not statistically significant, suggests its relationship with the degree of microvascular damage, and thus indirectly with endothelial dysfunction. An interesting result in the same regression model as above is finding a significant relationship between hsCRP and endothelin-1 (*R*² = 0.25, *p* = 0.049). 

Therefore, our results suggest the participation of endothelin-1 in the pathomechanism of RP but also its possible proinflammatory effect and endothelial activation. Proinflammatory and immunomodulating effects ET-1 on endothelium are confirmed by the results of research in other diseases [42]. 

Another parameter arousing interest as an endothelial dysfunction marker is sE-selectin. It is expressed on activated endothelial cells due to the effect of proinflammatory cytokines, and the current theory of atherosclerosis development ascribes to it an important role at the earliest stage of atheromatous plaque formation [43,44] At the same time, sE-selectin participation in the pathogenesis of Raynaud’s phenomenon is taken into account, since an increased sE-selectin expression is observed on the endothelial cells in the group of patients with Raynaud’s phenomenon, in whom SSc developed [12,45]. The results of our studies show a significantly higher sE-selectin concentration in the group of patients with secondary Raynaud’s phenomenon, in which “scleroderma pattern” of capillaries predominate. The relationship between abnormal capillaroscopic parameters and possible endothelial dysfunction was suggested by the analysis of multifactorial linear regression, which demonstrated that megacapillaries and microhemorrhages on capillaroscopy examination were the parameters most strongly determining the sE-selectin concentration in the whole study group (*R*^2^ = 0.21, *p* = 0.04).

These results are in concordance with the study by Valentini G et al., who demonstrated a higher prevalence of digital ulcers and increased serum levels of soluble E-selectin in RP patients, who had capillaroscopic “scleroderma pattern” on NVC, regardless of marker autoantibodies [46]. The researchers also connect the presence of the markers of inflammatory cell activation in patients demonstrating abnormalities on NVC with preclinical cardiopulmonary alterations [34,47]. Yamane et al. have found that soluble E-selectin concentration was significantly higher in patients with scleroderma and was positively correlated with the size of skin lesions and the extent of areas involved [48]. Luskiewicz-Potemska et al. demonstrated a significantly higher serum sE-selectin level in children and adolescents with RP and microangiopathy on NVC compared with those without microangiopathic complications with normal NVC pattern, which also can suggest subclinical vasculitis [26].

It should be emphasized that only children and adolescents with RP with abnormal capillaroscopic patterns (non-specific or scleroderma pattern) and low values of inflammatory process parameters, in whom the history is taken and clinical examination ruled out the classic cardiovascular disease risk factors, were qualified for the study presented, aimed at assessing microangiopathy with possible damage to the endothelial function.

Thus, it seemed interesting to answer the question whether microangiopathy in the studied patients is accompanied by the so-called low-grade inflammatory state, the recognized parameter of which is the hsCRP protein.

Nielen et al. presented an interesting analysis of a group of patients, blood donors, in whom they determined CRP level before rheumatoid arthritis (RA) development. They observed a higher CRP level in that group, even before the appearance of disease manifestations, compared with the control group [49]. The studies by other authors confirmed the importance of hsCRP in the development of the first cardiovascular episode [22,50,51]. Attention has also been paid to the participation of CRP in the inflammation accompanying atherosclerosis, which is related to its effect activating the endothelial cells to express adhesion molecules (soluble intercellular adhesion molecule 1 - ICAM-1, vascular cell adhesion molecule 1—VCAM-1, and E-selectin) [51,52,53,54]. In our studies, serum CRP concentration determined by high sensitivity method (hsCRP) was a strong determinant of the presence of avascular areas (*R*^2^ = 0.25, *p* = 0.03), giant and hemorrhages (*p* = 0.03). It seems very likely that the consequences of even a low-grade inflammatory state of peripheral microcirculation vessels include both functional and structural microangiopathy, which has been proven in the studies on children with other diseases [37,49].

The literature data and previous own studies also showed higher hsCRP concentration in secondary RP compared to primary RP, which may be associated with subclinical vasculitis and possible endothelial dysfunction [54,55]

It suggests a relationship between elevated CRP concentrations and the presence of abnormal capillaroscopic patterns in the aspect of subclinical microcirculatory inflammation process in patients with long-lasting RP [53]. 

Therefore, the importance of biochemical markers of endothelial injury and hsCRP in patients with long-term Raynaud’s syndrome and capillary microscopic features of microangiopathy can be considered in the context of precocious preclinical atherosclerosis [56]. Studies on the population of adult patients with CTD with or without RP clearly indicate an increased risk of cardiovascular disease in comparison to the healthy population [22,23,57]. In light of current knowledge, endothelial dysfunction is the key and the earliest factor, both for immunological vasculitis and atherosclerosis [58]. It seems, therefore, that microangiopathy of the "scleroderma pattern" present in the capillaroscopic examination in patients with Raynaud’s syndrome may be the result of endothelial damage—a common link connecting subclinical vasculitis and atherosclerosis. Undoubtedly, the impact on the risk of premature atherosclerosis in patients with CTDs has an atherogenic lipid profile, characterized by an increase in LDL, and triglycerides, and HDL reduction [51]. In our study, we found a higher total cholesterol concentration in both RP groups and a significantly lower concentration of HDL-cholesterol compared to the control group (*p* = 0.001). Researchers suggest that the anti-inflammatory/protective role of HDL may be in part related to their inhibitory effect on the expression of endothelial adhesion particles [51]. On the other hand, in the multivariate regression analysis, both the total cholesterol and LDL were determined by the presence of avascular areas. Because avascular fields are a "scleroderma pattern" image, they can indirectly indicate damage to the vascular endothelium. It has been proven that endothelial dysfunction is associated with increased permeability of the vascular wall to lipoproteins already at the early stage of atherosclerosis [59]. 

In summary, despite the limitations of interpretation of the results of our studies, NVC should be regarded as a useful microcirculatory diagnostic method not only in the context of CTDs with or without RP, but also in less typical indications, particularly in vascular diseases. 

In combination with the determination of biomarker of endothelial dysfunction, NVC may be considered as a putative marker for subclinical vasculitis and endothelial dysfunction as well as a candidate for a surrogate marker of preclinical stage of atherosclerosis. This, however, requires further extended studies.

## Figures and Tables

**Figure 1 jcm-08-00567-f001:**
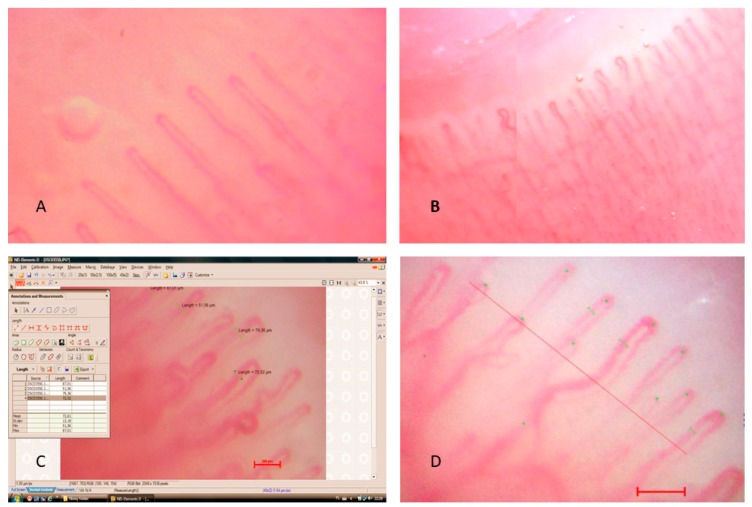
Capillaroscopic pictures of (**A**,**B**) normal capillary loops, (**C**,**D**) capillary loops in an example of NIS-ELEMENTS D2.30 computer program window.

**Figure 2 jcm-08-00567-f002:**
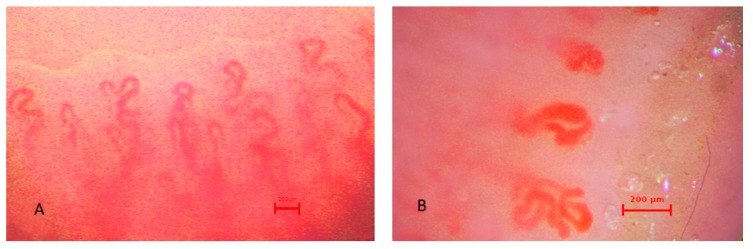
Capillaroscopic pictures of (**A**) tortuous capillaries, (**B**) meandering capillaries, giant capillaries, (**C**) spastic loops, aggregated flow, (**D**) irregular arrangement, giant capillaries, in the group with uRP.

**Figure 3 jcm-08-00567-f003:**
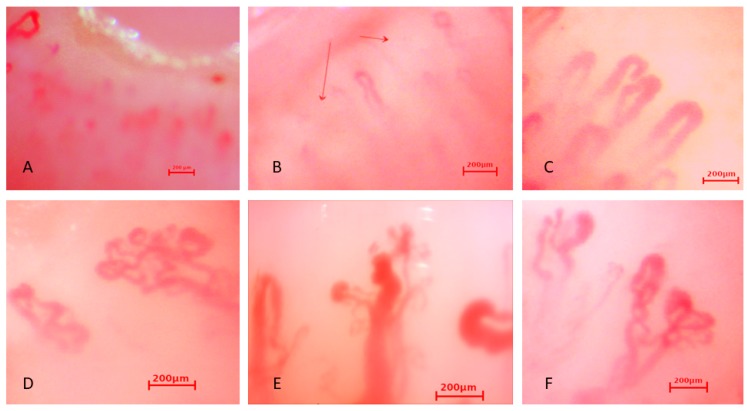
Capillaroscopic pictures of (**A**,**B**) reduced number of capillaries, (**C**) giant capillaries, (**D**,**F**) bizarre capillaries, (**E**) neoangiogenesis in the group with sRP.

**Figure 4 jcm-08-00567-f004:**
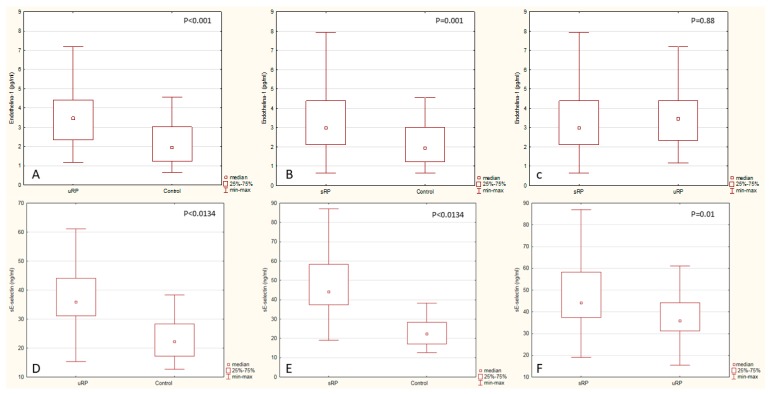
(**A**) Comparison of endothelin-1 concentration in the uRP group vs. endothelin-1 concentration in the control group (*p* < 0.001). (**B**) Comparison of endothelin-1 concentration in the sRP group vs. endothelin-1 concentration in the control group (*p* = 0.001). (**C**) Comparison of endothelin-1 concentration in the sRP group vs. endothelin-1 concentration in the uRP group (*p* = 0.88). (**D**) comparison of sE-selectin concentration in the uRP group vs. sE-selectin concentration in the control group (*p* < 0.0134). (**E**) comparison of sE-selectin concentration in the sRP group vs. sE-selectin concentration in the control group (*p* < 0.0134). (**F**) comparison of sE-selectin concentration in the sRP group vs. sE-selectin concentration in the uRP group (*p* = 0.01).

**Table 1 jcm-08-00567-t001:** Qualitative capillary parameters in the group of patients with Raynaud’s phenomenon (RP) and in the control group.

Capillary Parameters	Group with RP *n =* 66		Control Group *n =* 20		Chi-Square Test
	*n*	%	*n*	%	
Reduced number of capillaries (≤6/mm linear)	16	24.24	0	0.00	***p* = 0.015**
Irregular arrangement	40	66.67	0	0.00	***p* < 0.001**
Tortuous capillaries	23	34.85	7	35.00	NS
Meandering capillaries	33	50.00	4	20.00	***p* = 0.018**
Avascular areas	11	16.67	0	0.00	NS
Giant capillaries	36	54.55	0	0.00	***p* = 0.008**
Hemorrhages	13	19.70	0	0.00	***p* = 0.031**
Spastic loops	16	24.24	5	25.00	NS
Neoangiogenesis	5	7.58	0	0.00	*p* = 0.205

Bold: *p* < 0.05.

**Table 2 jcm-08-00567-t002:** Qualitative capillary parameters in the group of patients with RP in relation to gender.

Capillaroscopic Parameters	Male *n =* 21		Female *n =* 45		Ch-Square Test
	*n*	%	*n*	%	
Reduced number of capillaries (<6/mm)	4	19.08	12	18.18	NS
Irregular arrangement	11	52.38	29	64.44	NS
Tortuous capillaries	6	28.57	17	37.78	NS
Meandering capillaries	6	28.57	27	60.00	***p* = 0.017**
Avascular areas	4	19.05	7	15.56	NS
Giant capillaries	7	33.33	29	64.44	***p* = 0.018**
Microhemorrhages	5	23.81	8	17.78	NS
Spastic loops	5	23.81	11	24.44	NS
Neoangiogenesis	3	10.90	2	4.28	***p* < 0.001**

Bold: *p* < 0.05.

**Table 3 jcm-08-00567-t003:** Qualitative capillary parameters in the group of patients with RP in relation to age.

Capillaroscopic Parameters	Age ≤ 15 Years *n* = 13		Age >15 Years *n =* 53		Chi-Square Test
	*n*	%	*n*	%	
Reduced number of capillaries (<6/mm)	3	23.08	13	24.53	NS
Irregular arrangement	4	30.77	36	67.92	***p* =0.014**
Tortuous capillaries	1	7.70	22	41.51	***p* = 0.022**
Meandering capillaries	2	15.38	31	58.49	***p* = 0.005**
Avascular areas	3	23.08	8	15.09	NS
Giant capillaries	3	23.08	33	62.26	***p* = 0.011**
Microhemorrhages	4	30.77	9	16.98	NS
Spastic loops	4	30.77	12	22.64	NS
Neoangiogenesis	1	7.70	4	7.55	NS

Bold: *p* < 0.05.

**Table 4 jcm-08-00567-t004:** Assessment of the qualitative capillaroscopic parameters in children and adolescents with secondary Raynaud’s phenomenon (sRP) compared with the control group.

Capillaroscopic Parameters	Group with sRP *n =* 32		Control Group *n =* 20		Chi-Square Test
	*n*	%	*n*	%	
Reduced number of capillaries (≤6/mm)	12	37.50	0	0.00	***p* = 0.002**
Irregular arrangement	22	67.58	0	0.00	***p* < 0.001**
Tortuous capillaries	12	37.50	7	35.00	NS
Meandering capillaries	12	37.50	2	10	***p* = 0.030**
Giant capillaries	17	53.13	0	0.00	***p* < 0.001**
Microhemorrhages	8	25.00	0	0.00	***p* = 0.015**
Avascular areas	11	34.38	0	0.00	***p* = 0.003**
Spastic loops	9	24.24	5	25.00	NS
Neoangiogenesis	3	9.38	0	0.00	NS

Bold: *p* < 0.05.

**Table 5 jcm-08-00567-t005:** Assessment of the qualitative capillaroscopic parameters in children and adolescents with undifferentiated Raynaud’s phenomenon (uRP) compared with the control group.

Capillaroscopic Parameters	Group with uRP *n =* 34		Control Group *n =* 20		Chi-Square Test
	*n*	%	*n*	%	
Reduced number of capillaries (≤6/mm)	4	11.76	0	0.00	NS
Irregular arrangement	18	52.94	0	0.00	*p* < 0001
Tortuous capillaries	11	32.35	7	35.00	NS
Meandering capillaries	10	29.41	3	15.00	NS
Avascular areas	0	0.00	0	0.00	-
Giant capillaries	19	55.88	0	0.00	*p* < 0.001
Microhemorrhages	5	14.71	0	0.00	NS
Spastic loops	7	20.59	5	25.00	NS
Neoangiogenesis	2	5.88	0	0.00	NS

Bold: *p* < 0.05.

**Table 6 jcm-08-00567-t006:** Concentrations of the selected endothelial dysfunction markers and lipid metabolism parameters in the patients with undifferentiated (uRP) and secondary Raynaud’s phenomenon (sRP) compared with the control group.

Feature/Parameter	Group with uRP (*n =* 34) Mean ± SD	Group with sRP (*n =* 32) Mean ± SD	Control Group (*n =* 20) Mean ± SD	ANOVA Kruskal-Wallis Test
Age, years	13.07 ± 3.85	15.29 ± 4.52	14.73 ± 3.10	NS
RP duration, years	4.63 ± 2.34	5.74 ± 4.25	-	NS
hsCRP, mg/dL	0.42 ± 0.43	0.59 ± 0.51	0.078 ± 0.03	***p* < 0.001**
Endothelin-1, pg/mL	4.67 ± 3.38	5.74 ± 3,74	2.14 ± 0.6	***p* = 0.0134**
sE-selectin, ng/mL	39.06 ± 10.98	47.51 ± 16.96	20.92 ± 5.87	***p* < 0.001**
Total cholesterol, mg/dL	159.62 ± 25.17	163.25 ± 27.12	143.4 ± 20.58	NS
TG, mg/dL	70.29 ± 25.52	73.19 ± 40.31	83.2 ± 22.95	NS
LDL, mg/dL	85.11 ± 20.55	85.13 ± 21.02	81.52 ± 17.25	NS
HDL, mg/dL	60.41 ± 14.15	63.66 ± 15.52	78.54 ± 18.35	***p* = 0.001**

Bold: *p* < 0.05.

**Table 7 jcm-08-00567-t007:** Concentrations of the selected endothelial dysfunction markers and lipid metabolism parameters in the patients with RP in relation to gender.

Biochemical Parameters	Male *n =* 21		Female *n =* 45		Chi-Square Test
	*n*	%	*n*	%	
Total cholesterol > 180 mg/dL	1	4.76	7	15.56	NS
LDL cholesterol > 115 mg/dL	14	66.67	16	35.56	***p* = 0.018**
HDL cholesterol ≤ 48 mg/dL	15	71.43	27	60.00	NS
Triglicerides > 110 mg/dL	5	23.81	18	40.00	NS
hsCRP ≥ 0.5 mg/dL	6	28.57	27	60.00	***p* = 0.017**
sEselectin ≥ 40 ng/mL	15	71.43	17	37.78	**p = 0.011**
Endothelin-1 ≥ 2.5 pg/mL	13	61.19	31	68.89	NS

Bold: *p* < 0.05.

**Table 8 jcm-08-00567-t008:** Concentrations of the selected endothelial dysfunction markers and lipid metabolism parameters in the patients with RP in relation to age.

Biochemical Parameters	Age ≤ 15 Years *n* = 13		Age >15 Years *n* = 53		Chi-Square Test
	*n*	%	*n*	%	
Total cholesterol > 180 mg/dL	2	15.38	6	11.32	NS
LDL cholesterol > 115 mg/dL	1	7.69	29	54.72	***p* = 0.002**
HDL cholesterol ≤ 48 mg/dL	3	23.08	39	73.58	**p < 0.001**
Triglycerides > 110 mg/dL	1	7.69	22	41.51	***p* = 0.022**
E - selectin ≥ 40 ng/mL	7	53.85	25	47.17	NS
Endothelin ≥ 2.5 pg/mL	5	38.46	39	73.58	***p* = 0.016**
hsCRP≥ 0.5 mg/dL	3	23.08	30	56.60	***p* = 0.030**

Bold: *p* < 0.05.

**Table 9 jcm-08-00567-t009:** Factors (capillaroscopic parameters) determining sE-selectin concentration in the study group (determination coefficient *R*^2^ = 0.21).

Variable	BETA	Standard Error BETA	*p*
Reduced number of capillaries (<6/mm)	0.45	0.25	0.08
Irregular arrangement	0.03	0.21	0.88
Tortuous capillaries	−0.06	0.15	0.71
Meandering capillaries	0.00	0.14	0.99
Presence of avascular areas	0.09	0.15	0.52
Giant capillaries/microhemorrhages	0.29	0.14	**0.04**
Spastic capillaries	−0.11	0.13	0.37
Neoangiogenesis	−0.08	0.13	0.55

Bold: *p* < 0.05.

**Table 10 jcm-08-00567-t010:** Factors (capillaroscopic parameters) determining high-sensitivity c-reactive protein (hsCRP) concentration in the study group (determination coefficient *R*^2^ = 0.25).

Variable	BETA	Standard Error BETA	*p*
Reduced number of capillaries (<6/mm)	0.45	0.25	0.08
Irregular pattern	0.04	0.17	0.81
Tortuous capillaries	−0.08	0.16	0.64
Meandering capillaries	0.29	0.16	0.07
Giant capillaries/microhemorrhages	0.20	0.13	**0.03**
Presence of avascular areas	0.32	0.14	**0.03**
Spastic capillaries	−0.11	0.13	0.37
Neoangiogenesis	−0.08	0.13	0.55

Bold: *p* < 0.05.

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
