# Peer review of "Microangiopathy in Naifold Videocapillaroscopy and Its Relations to sE- Selectin, Endothelin-1, and hsCRP as Putative Endothelium Dysfunction Markers among Adolescents with Raynaud’s Phenomenon"

_jcm, 2019, doi:10.3390/jcm8050567_

Reviewer 1 Report

Stanilas et al present a paper exporing the impact of Naifold Videocapillaroscopy to asess Raynaud Syndrome. The results are novel and interesting.

Major:

Introduction: please describe briefly the method "nailfold videocapillaroscopy"

Patients "did they sign a consent form ? What is the number of ethics commitee agreement?
Can you check the effect of the age of the children on the results, if any, please?

Were the experimentators blinded to the children's condition?

I would suggest to club all these little figures into one big figure.

Line 305 "The literature data and previous own studies" References are needed.

Minor :

Abstract: correct "asses"
Abstract line 26: add a punctuation

Lines 48-50 rephrase "So, the hypothesis suggesting that in patients with many years of history of paroxysmal RP with 48 or without CTDs and structural abnormalities of the smallest blood vessels (capillaries) found by 49 means of NVC can be a preclinical sign of endothelial damage and early phase of vasculitis in micro- 50 and macrocirculation seems interesting"

line54 : correct "mediate in the"

Line 80: correct "systemic blood vessel abnormalitiesPatients and methods"

Line 121 rephrase "on an empty stomach, during"

line 223 correct "biochemical parameters shown significant"

line 235  correct "gropu"

line 241-42 rephrase "were more frequent 241 significantly different in patients"

line 244 rephrase "and limited data on among this age group,"

line 256, rephrase "as above is 256 the finding a significant relationship between"

Author Response

General: Thank you very much for you extremely valuable and inspiring remarks!

 Major:

Introduction: please describe briefly the method "nailfold videocapillaroscopy"

Done, lines 52-57

Patients "did they sign a consent form ? What is the number of ethics commitee agreement?
Can you check the effect of the age of the children on the results, if any, please?

 Thank you very much for this remarks and advices

The patients signed the consent form, which was necessary to obtain the consent of the Bioethics Committee - the approval number of the Bioethics Committee was entered in the manuscript – line 213.

Tables with capillary and biochemical parameters have been added, broken down by age. Thank you for this advice, some significant differences appeared described in the results section and the discussion.

 According to the second reviewer's suggestion, the capillary and biochemical parameters were analyzed in relation to the gender, as well. The age criteria ≤15 and> 15 years was admitted conventionally, but it is reliable, because according to pediatric endocrinologists 99.0% of the population aged 15 years is in the late maturation phase.
Based on the published capillaroscopic studies in healthy children, both the number of capillaries and the capillary system stabilize during puberty.

Were the experimentators blinded to the children's condition?

Thank you very much for thisimportant question. We assumed that the qualitative assessment of capillaroscopic parameters by an experienced pediatric rheumatologist, with more than 20 years of practice in performing NVC in children and adolescents - although known to the respondents - will be more reliable than the assessment of a less experienced researcher.
For the purpose of this work, the quality parameters were additionally analyzed on the capillaries recorded by the same researcher in the computer (AG). The second researcher (MB), who did not know the patients’status, performed the measurements of the number of capillaries.

We can assume, then, that it was partially blinded. Some more explanation was included in the Method section, line 139.

I would suggest to club all these little figures into one big figure.

Done, thank you very much, starting with the line 281.

Line 305 "The literature data and previous own studies" References are needed.

Done, thank you very much.

Minor :

Abstract: correct "asses"
Abstract line 26: add a punctuation

Lines 48-50 rephrase "So, the hypothesis suggesting that in patients with many years of history of paroxysmal RP with 48 or without CTDs and structural abnormalities of the smallest blood vessels (capillaries) found by 49 means of NVC can be a preclinical sign of endothelial damage and early phase of vasculitis in micro- 50 and macrocirculation seems interesting"

line54 : correct "mediate in the"

Line 80: correct "systemic blood vessel abnormalitiesPatients and methods"

Line 121 rephrase "on an empty stomach, during"

line 223 correct "biochemical parameters shown significant"

line 235  correct "gropu"

line 241-42 rephrase "were more frequent 241 significantly different in patients"

line 244 rephrase "and limited data on among this age group,"

line 256, rephrase "as above is 256 the finding a significant relationship between"

 We asked a help from native speaker specializing in medical translations with these suggestion, we hope it is good enough now. Thank you very much!

 Reviewer 2 Report

TO THE AUTHORS:

Journal of Clinical Medicine; jcm-469225

Title: Microangiopathy in Nailfold Videocapillaroscopy and its Relations to sE- Selectin, Endothelin-1, and hsCRP as Putative Endothelium Dysfunction Markers among Adolescents with Raynaud's Phenomenon

Stanislaw et al examined the structure of the nailfold microcirculation and blood levels of endothelial and lipid biomarkers of young subjects (6-19 yrs) with and without Raynaud’s Syndrome or Phenomenon (RS or RP; undifferentiated & secondary). Specific conclusions are not apparent but the authors generally point to the combination of nailfold videocapillaroscopy (NVC) and measurement of endothelial biomarkers as a useful diagnostic method for autoimmune connective tissue diseases in addition to vascular diseases for atherosclerosis. While I appreciate the clinical potential of the authors’ study and hope it ultimately becomes successful in accord with their intentions, the current manuscript is in need of vast improvement before consideration for publication. See comments below.

Comments:

(1) Lines 48-51 (Introduction); Hypothesis.

It took several reads to understand the hypothesis statement. I recommend rephrasing it as two sentences (e.g., sentence 1 focuses on connecting RP with a structurally/functionally remodeled microcirculation & sentence 2 on how NVC combined with blood biomarker measurements can be used as a diagnostic tool for identifying preclinical prognosticators of endothelial dysfunction.

(2) Lines 56-80 (Introduction) & Lines 246-316 (Discussion); Rationale/discussion of chosen biomarkers

While there is no dispute with the importance of the selected biomarkers for the study, there are several others that are also representative of endothelial structure/function and not adequately addressed in the manuscript (VE-cadherin, eNOS, VEGF, etc.). In the Introduction, it is recommended that the authors more completely establish the rationale of studying their particular set of biomarkers out of many. It would seem most ideal to first clearly describe the potential interactions between nailfold microcirculatory structural changes, lipid profiles, and select endothelial biomarkers and then clarify representative markers/indicators of each type of experimental variable to be studied and why.

In general for both Introduction and Discussion, the information provided appears to be scattered without a logical flow or transitioning within and between paragraphs. A key example is shown in Lines 51-55 mentioning reactive oxygen species (ROS) signaling. How does this statement follow from Lines 48-51 (hypothesis) and how does this set up reading/thinking for the next paragraph (Lines 56-60; potential role of endothelin-1 & E-selectin)? Also, the authors don’t even examine or discuss ROS levels/signaling after this point in the manuscript.

(3) Lines 90-91 & Lines 93-95; Human Subject composition

Instead of bundled as total numbers for RP vs. control, the composition of gender and age should be clarified for each type of individual RP and control group.

(4) Lines 110-119 & Results; NVC measurements

It is strongly recommended that the authors show representative examples (i.e., images) of the different vascular patterns while visually indicating the precise capillaroscopic variables that were measured and how.

(5) Tables/Figures.

Can the authors report comparisons of males vs. females as well?

(6) Figures/Figure Legends

There is no Figure 1 and the first figure starts as Figure 2. Figures 2 through 4 in particular have an excessive scale range (about 2-fold greater than necessary) that visually diminishes the size of range bars. The Figure Legends do not adequately indicate basic information such as n-values and the specification of data shown (e.g., mean or median, SE or SD, confidence interval). The Statistical Analysis section is somewhat incomplete in this regard as well.

(7) Entire manuscript

There are several typos (e.g., Line 80) and undefined abbreviations (e.g., JIA, HDD, IBL, TMD) throughout the manuscript.

Author Response

Journal of Clinical Medicine; jcm-469225

Title: Microangiopathy in Nailfold Videocapillaroscopy and its Relations to sE- Selectin, Endothelin-1, and hsCRP as Putative Endothelium Dysfunction Markers among Adolescents with Raynaud's Phenomenon

Stanislaw et al examined the structure of the nailfold microcirculation and blood levels of endothelial and lipid biomarkers of young subjects (6-19 yrs) with and without Raynaud’s Syndrome or Phenomenon (RS or RP; undifferentiated & secondary). Specific conclusions are not apparent but the authors generally point to the combination of nailfold videocapillaroscopy (NVC) and measurement of endothelial biomarkers as a useful diagnostic method for autoimmune connective tissue diseases in addition to vascular diseases for atherosclerosis. While I appreciate the clinical potential of the authors’ study and hope it ultimately becomes successful in accord with their intentions, the current manuscript is in need of vast improvement before consideration for publication. See comments below.

Comments:

General: Thank you very much for you extremely valuable and inspiring remarks!

(1) Lines 48-51 (Introduction); Hypothesis.

It took several reads to understand the hypothesis statement. I recommend rephrasing it as two sentences (e.g., sentence 1 focuses on connecting RP with a structurally/functionally remodeled microcirculation & sentence 2 on how NVC combined with blood biomarker measurements can be used as a diagnostic tool for identifying preclinical prognosticators of endothelial dysfunction.

Thank you very much. Indeed, it needed clarification. I hope it is easier to understand now – we tried to follow your suggestion about splitting into 2 sentences and  asked a native speaker to help us with this passage, currently lines 67-70

 (2) Lines 56-80 (Introduction) & Lines 246-316 (Discussion); Rationale/discussion of chosen biomarkers

While there is no dispute with the importance of the selected biomarkers for the study, there are several others that are also representative of endothelial structure/function and not adequately addressed in the manuscript (VE-cadherin, eNOS, VEGF, etc.). In the Introduction, it is recommended that the authors more completely establish the rationale of studying their particular set of biomarkers out of many. It would seem most ideal to first clearly describe the potential interactions between nailfold microcirculatory structural changes, lipid profiles, and select endothelial biomarkers and then clarify representative markers/indicators of each type of experimental variable to be studied and why.

Thank you very much for this advice. The introduction and the discussion have been reformulated taking into account you valuable comments. We explained more indepth the rationale of studying the particular set of biomarkers out of the others. We also described the potential interactions between nailfold microcirculatory structural changes, lipid profiles, and select endothelial biomarkers in the perspective of endothelium damage and potential link to preclinical atherosclerosis

 In general for both Introduction and Discussion, the information provided appears to be scattered without a logical flow or transitioning within and between paragraphs. A key example is shown in Lines 51-55 mentioning reactive oxygen species (ROS) signaling. How does this statement follow from Lines 48-51 (hypothesis) and how does this set up reading/thinking for the next paragraph (Lines 56-60; potential role of endothelin-1 & E-selectin)? Also, the authors don’t even examine or discuss ROS levels/signaling after this point in the manuscript.

Thank you very much for this remark. The introduction and the discussion have been quite extensively reformulated taking into account you valuable comments, hopefully, the logical flow and or transitioning within and between paragraphs is much improved. We removed statements which are not directly linked to the study hypothesis, aims and methodology.

 (3) Lines 90-91 & Lines 93-95; Human Subject composition

Instead of bundled as total numbers for RP vs. control, the composition of gender and age should be clarified for each type of individual RP and control group.

Done according you advice, thank you very much! Currently lines 123-128

(4) Lines 110-119 & Results; NVC measurements

It is strongly recommended that the authors show representative examples (i.e., images) of the different vascular patterns while visually indicating the precise capillaroscopic variables that were measured and how.

Done according you advice, thank you very much! Capillaries photographs were added, of the different vascular patterns while visually indicating the precise capillaroscopic variables that were assessed, with a view into the programme supporting the assessment in one of the examples photos 1-3, starting with the line 159

  (5) Tables/Figures.

Can the authors report comparisons of males vs. females as well?

Tables with capillary and biochemical parameters have been added, broken down by gender. According to the second reviewer's suggestion, the capillary and biochemical parameters were analysed also in relation to the age. Thank you for this advice, some significant differences appeared described in the results section and the discussion.

 (6) Figures/Figure Legends

There is no Figure 1 and the first figure starts as Figure 2. Figures 2 through 4 in particular have an excessive scale range (about 2-fold greater than necessary) that visually diminishes the size of range bars. The Figure Legends do not adequately indicate basic information such as n-values and the specification of data shown (e.g., mean or median, SE or SD, confidence interval). The Statistical Analysis section is somewhat incomplete in this regard as well.

Thanks a lot. The numbering of charts has been improved and the legend of the charts has been supplemented with an extended description of statistic parameters, starting with line 281

(7) Entire manuscript

There are several typos (e.g., Line 80) and undefined abbreviations (e.g., JIA, HDD, IBL, TMD) throughout the manuscript.

The typos were corrected and abbreviations explained, or removed, thank you very much!

Round  2

Reviewer 2 Report

TO THE AUTHORS:

Journal of Clinical Medicine; jcm-469225

Title: Microangiopathy in Nailfold Videocapillaroscopy and its Relations to sE- Selectin, Endothelin-1, and hsCRP as Putative Endothelium Dysfunction Markers among Adolescents with Raynaud's Phenomenon

Stanislaw et al examined the structure of the nailfold microcirculation and blood levels of endothelial and lipid biomarkers of young subjects (6-19 yrs) with and without Raynaud’s Syndrome or Phenomenon (RS or RP; undifferentiated & secondary). The authors refine the use of a combination of nailfold videocapillaroscopy (NVC) and measurement of endothelial biomarkers as a useful microcirculatory diagnostic method for autoimmune connective tissue diseases in addition to vascular diseases for atherosclerosis. In general, the authors’ were very responsive to my previous comments. At this stage, the manuscript needs some final proofreading for undefined terms (e.g., define SSc) and typos while ensuring that scale bars are indicated across the recently added images/photos.

Author Response

Dear Reviewer,
Thank you very much again for your valuable comments and recommendations: 

(At this stage, the manuscript needs some final proofreading for undefined terms (e.g., define SSc) and typos while ensuring that scale bars are indicated across the recently added images/photos.)

We added definitions to the undefined terms (SSc, ICAM-1, VCAM-1), checked and corrected several typos. We also added scale bars to the added photos.
Thank you very much again and kindest regards!